A non-cardiomyocyte autonomous mechanism of cardioprotection involving the SLO1 BK channel

Wojtovich Andrew P. 1
Nadtochiy Sergiy M. 2
Urciuoli William R. 2
Smith Charles O. 3
Grunnet Morten 4 5
Nehrke Keith 1
Brookes Paul S. 2 paul_brookes@urmc.rochester.edu
1 Department of Medicine, University of Rochester Medical Center , Rochester, NY , USA
2 Department of Anesthesiology, University of Rochester Medical Center , Rochester, NY , USA
3 Department of Biochemistry, University of Rochester Medical Center , Rochester, NY , USA
4 Lundbeck AS , Valby , Denmark
5 Acesion Pharma , Copenhagen , Denmark
Xie Lai-Hua
Electronic publication date: 2013 Mar 5
Publication date: 2013
Volume: 1
Electronic Location ID: e48
Received 2013 Jan 3; Accepted 2013 Feb 19
Copyright: © 2013 Wojtovich et al.
Copyright year: 2013
Copyright holder: Wojtovich et al.
License: This is an open access article distributed under the terms of the Creative Commons Attribution License, which permits unrestricted use, distribution, and reproduction in any medium, provided the original author and source are credited.
License URL: https://creativecommons.org/licenses/by/3.0/

Keywords: Large conductance potassium channel, Ischemia, Reperfusion, Preconditioning, Cardiac neurons, NS1619, NS11021

Funding: National Institutes of Health RO1-HL071158 RO1-GM087483 American Heart Association 11POST7290028 Work was funded by grants from the US National Institutes of Health, HL071158 (to PSB) and GM087483 (to PSB & KWN). APW is funded by an American Heart Association Founder’s Affiliate Postdoctoral Fellowship award (11POST7290028). The funders had no role in study design, data collection and analysis, decision to publish, or preparation of the manuscript.

==============================
Opening of BK-type Ca2+ activated K+ channels protects the heart against ischemia-reperfusion (IR) injury. However, the location of BK channels responsible for cardioprotection is debated. Herein we confirmed that openers of the SLO1 BK channel, NS1619 and NS11021, were protective in a mouse perfused heart model of IR injury. As anticipated, deletion of the Slo1 gene blocked this protection. However, in an isolated cardiomyocyte model of IR injury, protection by NS1619 and NS11021 was insensitive to Slo1 deletion. These data suggest that protection in intact hearts occurs by a non-cardiomyocyte autonomous, SLO1-dependent, mechanism. In this regard, an in-situ assay of intrinsic cardiac neuronal function (tachycardic response to nicotine) revealed that NS1619 preserved cardiac neurons following IR injury. Furthermore, blockade of synaptic transmission by hexamethonium suppressed cardioprotection by NS1619 in intact hearts. These results suggest that opening SLO1 protects the heart during IR injury, via a mechanism that involves intrinsic cardiac neurons. Cardiac neuronal ion channels may be useful therapeutic targets for eliciting cardioprotection.

Introduction

All tissues, including the heart, possess endogenous mechanisms to protect against ischemia-reperfusion (IR) injury, and the opening of K+ channels is central to many cardioprotective paradigms including ischemic preconditioning (IPC) and anesthetic preconditioning (APC) (Facundo, Fornazari & Kowaltowski, 2006). The mitochondrial KATP channel is perhaps the most widely associated with cardioprotection (O’Rourke, 2007; Zoratti et al., 2009), but a large-conductance (BK) channel encoded by the Slo gene family is also proposed to play a protective role in the heart  (Bentzen et al., 2009; Wojtovich et al., 2011). The Slo encoded channels comprise 3 isoforms (SLO1/ 2/3) (Salkoff et al., 2006), and although a channel resembling SLO1 (KCa1.1, KCNMA1) has been reported in mitochondria (Siemen et al., 1999), gene deletion experiments show that Slo1 is not required for cardioprotection by IPC or APC (Wojtovich et al., 2011).

Despite ongoing debate regarding the identity of a mitochondrial BK channel, it is widely reported that SLO1 is expressed in the heart (Chen et al., 2005; Jiang et al., 1999; Tseng-Crank et al., 1994). Furthermore, SLO1 activation by small molecules such as NS1619 and NS11021 is cardioprotective (Xu et al., 2002; Bentzen et al., 2007). SLO1 is not found at the cardiomyocyte plasma membrane (Xu et al., 2002), but SLO1 channel activity has been reported in intra-cardiac neurons (Franciolini et al., 2001) and Purkinje fibers (Callewaert, Vereecke & Carmeliet, 1986), where it is thought to play a role in regulating heart rate (Imlach et al., 2010).

Despite the assumption that NS1619 and NS11021 protect via a (mitochondrial) SLO1 (Xu et al., 2002), many non-specific effects such as opening of other ion channels (Park et al., 2007; Saleh et al., 2007; Holland et al., 1996) and inhibition of mitochondrial function (Aldakkak et al., 2010; Aon et al., 2010; Cancherini, Queliconi & Kowaltowski, 2007; Debska et al., 2003; Kicinska & Szewczyk, 2004) have been reported for these compounds, and a pharmaco-genomic study has not been conducted. Such SLO1-independent effects could underlie the cardioprotective benefit of these compounds (Cancherini, Queliconi & Kowaltowski, 2007; Kicinska & Szewczyk, 2004), and herein we sought to investigate the contribution of SLO1 to the cardioprotection elicited by NS1619 and NS11021, including studies in Slo1−/− mice to control for off-target effects. We report on the discovery of a non-cardiomyocyte autonomous mechanism of cardioprotection involving SLO1 and intrinsic cardiac neurons.

Materials & Methods

Reagents, animals

NS11021 was a gift from Neurosearch A/S (Ballerup, Denmark). All other chemicals were of the highest grade available from Sigma (St. Louis, MO) unless otherwise specified. Slo1−/− (KCNMA1−/−) mice were generated on the FVB background via deletion of Exon1, which contains the translation start site and the S0 trans-membrane segment (Meredith et al., 2004). This results in both transcripts and protein being undetectable, and it has been previously shown that the lack of an S0 domain results in a nonfunctional BK channel  (Zarei et al., 2001; Zarei et al., 2004). Animals were genotyped by tail biopsy PCR, as previously described (Meredith et al., 2004). All experiments used male WT and Slo1−/− littermates age 6–8 weeks. Mice were maintained in an AAALAC-accredited pathogen-free barrier facility with food and water available ad libitum. All procedures were in accordance with the NIH Guide for the Care and Use of Laboratory Animals (2011 revision) and were approved by the University Committee on Animal Resources.

Mouse Langendorff ex-vivo perfused heart

Following avertin anesthesia, a thoracotomy was performed, the aorta cannulated in-situ and the heart rapidly transferred to perfusion apparatus as previously described (Wojtovich et al., 2011; Nadtochiy et al., 2009). The heart was perfused with Krebs-Henseleit buffer in constant flow mode (4 ml/min). The following experimental protocols were observed: (i) IR injury: 20 min normoxic perfusion, 40 min global ischemia, 60 min reperfusion. (ii) NS1619+IR: 10 min normoxic perfusion, 10 min of 5 µM NS1619, 30 s washout, then IR as above. (iii) NS11021+IR: 10 min normoxic perfusion, 10 min of 500 nM NS11021, 30 s washout, then IR as above.

Studies involving hexamethonium were independently controlled and the following experimental protocols were observed: (iv) IR injury: 25 min normoxic perfusion, 35 min global ischemia, 60 min reperfusion. (v) NS1619+IR: 15 min normoxic perfusion, 10 min of 5 µM NS1619, 30 s washout, then IR as above. (vi) NS1619+IR+Hexamethonium: 12.5 min normoxic perfusion, 2.5 min of 500 µM hexamethonium, 10 min of 5 µM NS1619 plus 500 µM hexamethonium, 30 s washout of NS1619 only, 35 min global ischemia, 5 min reperfusion plus 500 µM hexamethonium, 55 min reperfusion. (vii) Atpenin A5 (AA5)+IR: 5 min normoxic perfusion, 20 min of 10 nM AA5, 30 s washout, then IR as above. (viii) AA5+IR+Hexamethonium: 2.5 min normoxic perfusion, 2.5 min of 500 µM hexamethonium, 20 min of 10 nM AA5 plus 500 µM hexamethonium, 30 s washout of AA5 only, 35 min global ischemia, 5 min reperfusion plus 500 µM hexamethonium, 55 min reperfusion. Note that different ischemic times (35 vs. 40 min) were used to ensure adequate independent controls for every group examined.

To assess neuronal survival, hearts were exposed to 100 µM nicotine for 5 min following the reperfusion period. Compounds were delivered via a syringe pump linked to an injection port immediately above the perfusion cannula. The volume of added solutions was ≤0.05% of the total perfusate volume. Following experimental protocols, hearts were TTC stained and imaged as previously described to quantify infarct size (Wojtovich et al., 2011; Nadtochiy et al., 2011).

Mouse cardiomyocyte isolation

Following avertin anesthesia, the aorta was cannulated in-situ and the heart rapidly transferred to perfusion apparatus. The heart was perfused for 3 min with a modified Krebs-Henseleit “buffer A” (118 mM NaCl, 4.7 mM KCl, 1.2 mM KH2PO4, 1.2 mM MgSO4, 10 mM HEPES, 24 mM NaHCO3, 5.5 mM glucose, bubbled with 95% O2, 5% CO2, pH 7.35, 37 °C) at 3 ml/min to remove blood. The heart was then perfused with digestion buffer (buffer A plus 0.25 mM CaCl2, 30 mM taurine, 10 mM 2,3-butanedione monoxime, 0.0278% Trypsin, 25 mg collagenase A, 75 mg collagenase D) at 3 ml/min for 12 min. The heart was removed and teased apart with forceps. Tissue was dispersed with 10 ml stop buffer (buffer A plus 12.5 µM CaCl2, 10% FBS) with a transfer pipette, then passed through a 200 µm filter. Myocytes were settled by gravity for 10 min, supernatant removed, and pellet resuspended in 10 ml stop buffer. Ca2+was gradually increased from 12.5 µM to 0.25 mM through four additions of CaCl2 over the course of 8 min. Myocytes were settled by gravity for 10 min, supernatant removed, and pellet resuspended in 5 ml normoxic buffer (buffer A plus 0.25 mM CaCl2, 10% FBS). Myocytes were counted and viability determined by Trypan blue exclusion. One mouse heart typically yielded 1.2 × 106 cells (75% viable) and cells were used for simulated IR injury immediately after isolation.

Mouse cardiomyocyte simulated IR injury

Myocytes were equilibrated at a concentration of 105 cells/ml for 10 min in normoxic buffer. Incubations were in 50 ml round-bottom tubes in a shaking water bath (80 cycles/min) at 37 °C. Myocytes were divided in to the following groups (i) Control, 100 min in normoxic buffer; (ii) Simulated ischemia-reperfusion (SIR), 10 min normoxic buffer, then 1 h ischemia buffer (118 mM NaCl, 4.7 mM KCl, 1.2 mM KH2PO4, 1.2 mM MgSO4, 10 mM HEPES, 24 mM NaHCO3, 0.25 mM CaCl2, 10% FBS, gassed with 95% N2, 5% CO2, pH 6.5, 37 °C), followed by 30 min normoxic buffer; (iii) Drug + SIR, comprising treatment with compounds 10 min prior to SIR in normoxic buffer, then 1 h ischemia buffer, 30 min normoxic buffer. Drugs tested (final concentrations in parentheses) were: NS1619 (5 µM), NS11021 (500 nM) and diazoxide (10 µM). Drugs were dissolved in DMSO and the vehicle comprised <0.1% of total incubation volume. DMSO was included in the control and SIR groups at the appropriate times. To change buffers, cells were centrifuged at 31 × g for 2 min, and the pellet resuspended in the appropriate buffer. At the end of protocols, viability was determined by Trypan blue exclusion.

Isolated mitochondrial BK channel activity

Mitochondria were isolated from 3 mouse hearts as previously described  (Wojtovich et al., 2010). The mitochondria were incubated with 20 µM BTC-AM and 0.05% Pluronic F-127 for 10 min at room temperature. The final mitochondrial pellet was suspended in 225 µl and stored on ice until use, within 1.5 h Tl+ uptake into mitochondria was measured using a Varian Cary Eclipse spectrofluorimeter as previously described (Wojtovich et al., 2010) by monitoring changes in BTC fluorescence (λEX 488 nm, λEM 525 nm). Where indicated, ATP (to block mKATP channel), NS compounds (to open mBK channels) and paxilline (to block BK channels) were present at concentrations indicated.

Total heart histochemical acetylcholinesterase staining

Following IR protocols as described above, hearts were perfused with 10 mM phosphate buffered saline (PBS; pH 7.4 at 25 °C) for 3 min at 4 ml/min. Control hearts not exposed to IR injury were cannulated and perfused with PBS. Hearts were then stained for acetylcholinesterase (Rysevaite et al., 2011). Hearts were prefixed with 4% formalin for 30 min at 4 °C, then washed with PBS and incubated in PBS supplemented with hyaluronidase (0.5 mg/100 ml) for 18 h at 4 °C. The hearts were then placed in Karnovsky-Roots medium (60 mM sodium acetate, 2 mM acetylthiocholine iodide, 15 mM sodium citrate, 3 mM CuSO4, 0.5 mM K3Fe(CN)6, 0.1% trition-X 100, 0.5 mg/100 ml hyaluronidase, pH 5.6 at 4 °C) for 3 h at 4 °C. Following staining, hearts were kept in 10% formalin and visualized under fluorescent illumination using GFP filters on an Olympus MVX wide field dissecting microscope. Images were obtained using a PixeLink camera. The fluorescent illumination provided increased contrast compared to visible light with the monochrome CCD.

Statistics

Data presented are means ± SEM. Statistical significance (p < 0.05) between multiple groups was determined using analysis of variance (ANOVA) with Bonferroni correction. Mixed effects models were generated using “R” software (R Development Core Team) and the R packages lme4 (Bates & Maechler, 2009) and languageR (Baayen, 2008; Baayen, 2009). As fixed effects, we included treatment conditions, genotype, and their interaction in the model. Normality and homogeneity were checked using visual inspections of plots of residuals against fitted values. Likelihood ratio tests comparing the models with fixed effects to the null models using only random effects, were used to assess the accuracy of the mixed effects models. The model was further analyzed using pair wise comparison t-test. We rejected results where the model including fixed effects did not differ significantly from the null model. For all mixed effects models we present MCMC-estimated p-values that are considered significant at the α < 0.05 level.

Results

Protection of intact hearts by NS1619 and NS11021 requires Slo1

Previously we showed that cardioprotection by both IPC and APC in intact hearts is SLO1 independent (Wojtovich et al., 2011). However, the SLO1 openers NS1619 and NS11021 are still reported to protect against IR injury in intact hearts  (Bentzen et al., 2009; Xu et al., 2002), suggesting that SLO1 alone can afford protection. To examine the pharmaco-genomics of SLO1, NS1619 and NS11021 in more detail, we utilized an ex-vivo perfused heart model of IR injury. Representative raw data tracings of cardiac function during IR injury are shown in Fig. 1. Hearts from WT and Slo1−/− mice exhibited similar sensitivity to IR (quantitative data shown in Fig. 2), and as previously reported both NS1619 and NS11021 protected the WT heart from IR injury (enhanced functional recovery and reduced infarct size), with NS11021 being 10-fold more potent than NS1619. As anticipated, protection was lost in Slo1−/− hearts, suggesting that SLO1 is a bona-fide target for the cardioprotective effects of NS1619 and NS11021 in the intact heart. While it is possible that SLO1 is merely a downstream mediator of protection by NS1619 and NS11021, Occam’s razor would suggest SLO1 is the actual target.

Figure 1 Representative left ventricular pressure traces from perfused hearts.

Traces are shown for wild-type (WT) and Slo1−/− hearts, in the presence of vehicle (Ctrl.), NS1619 or NS11021, as per the methods. Traces are compressed on the time (x) axis. The top and bottom boundaries of the black shaded area represent the systolic and diastolic pressures, respectively. The onset of ischemia (I) and reperfusion (R) are indicated by arrows. Scale bars on each trace represent 10 min. (x-axis) and 50 mmHg (y-axis).

Figure 2 SLO1 dependent protection of perfused mouse heart against ischemia-reperfusion (IR) injury.

Perfused FVB littermate WT (white symbols) and Slo1−/− (gray symbols) mouse hearts were subjected to IR injury. Where indicated, hearts were treated with NS1619 (5 µM) or NS11021 (500 nM) prior to ischemia. (A): Left-ventricular function was determined as rate pressure product (RPP; heart rate × left ventricular developed pressure) and expressed as percent of initial value. Statistical significance between groups was determined using a mixed-effects model. *p < 0.05 between NS11021 and control (IR alone), #p < 0.05 between NS1619 and control (IR alone). (B): Following IR protocols, hearts were sliced, stained with TTC and fixed to delineate live (red) from dead or infarcted tissue (white). Infarcts were quantified by planimetry and expressed as a percent of the area at risk (100% in this global ischemia model). Images above the graph show representative slices from each group. Within each group, individual values are on the left, and means ± SEM on the right (N = 7 for IR group and 6 for all other groups). *p < 0.05 between NS11021 and control (IR alone), #p < 0.05 between NS1619 and control (IR alone).

Protection of isolated cardiomyocytes by NS1619 and NS11021 is Slo1 independent

Slo1 mRNA is expressed at low levels in whole heart (Chen et al., 2005; Jiang et al., 1999; Tseng-Crank et al., 1994), and it has been suggested that SLO1 is present in cardiomyocyte mitochondria (Xu et al., 2002). We therefore hypothesized that SLO1 in cardiomyocyte mitochondria may play a role in mediating the protective effects of NS1619 and NS11021. To test this hypothesis, adult mouse ventricular cardiomyocytes were isolated from WT and Slo1−/− hearts and subjected to simulated IR injury. As expected, both NS1619 and NS11021 elicited cytoprotection (reduced cell death) in response to IR (Fig. 3). However, this protection was present in both WT and Slo1−/− derived myocytes, indicating it did not require SLO1 activity. As a control, the mitochondrial KATP opener diazoxide was protective in both WT and Slo1−/− cells. Overall these data suggest that the protection of cardiomyocytes by NS1619 and NS11021 may be due to off-target (non-SLO1) effects of these molecules, as reported elsewhere (Park et al., 2007; Saleh et al., 2007; Holland et al., 1996; Aldakkak et al., 2010; Aon et al., 2010; Cancherini, Queliconi & Kowaltowski, 2007; Debska et al., 2003; Kicinska & Szewczyk, 2004).

Figure 3 SLO1 independent protection of adult mouse cardiomyocytes against simulated ischemia-reperfusion (IR) injury.

Myocytes were isolated from adult littermate WT (white) and Slo1−/− (gray) mouse hearts, and used immediately in the model of IR injury. Where indicated, cells were treated with NS1619 (5 µM), NS11021 (500 nM) or diazoxide (DZX, 10 µM). Upon completion of the IR protocol, cell viability was measured via Trypan blue exclusion and expressed as a percent of control (viability in control normoxic groups was: WT 70 ± 3%, Slo1−/− 69 ± 4%). Experimental conditions are listed below the x-axis. Data are means ± SEM, with N for each group listed in parentheses at the base off each bar. Each N represents an independent cardiomyocyte preparation. *p < 0.05 vs. IR alone (ANOVA).

NS1619 and NS11021 exhibit Slo1 independent effects on isolated mitochondria

To directly test whether NS1619 and NS11021 were able to activate BK channels in isolated mitochondria, a modified thallium (Tl+) flux assay was utilized (Wojtovich et al., 2010). As expected, both compounds were able to elicit a Tl+ flux, in a manner sensitive to the BK channel blocker paxilline (Fig. 4). However, this flux was evident in mitochondria derived from both WT and Slo1−/− hearts, indicating it did not originate at the level of SLO1. Furthermore, bona fide mitochondrial K+ channels exhibit a slowing of Tl+ flux kinetics by increasing concentrations of K+ and not Na+ (since only K+ competes for Tl+ uptake). However, such competition was not seen for NS induced Tl+ flux (data not shown), suggesting it is not ion-specific. Together these data support the notion that NS compounds have non-specific effects on mitochondria. Such effects may account for the non Slo1-dependent protection observed in isolated cardiomyocytes (Fig. 3).

Figure 4 SLO1 independent K+ channel activity in isolated mouse heart mitochondria.

Mitochondria were isolated from littermate WT (white) and Slo1−/− (gray) mouse hearts and loaded with Tl+-sensitive fluorophore. Mitochondrial K+ channel activity in the presence of NS compounds was determined using the Tl+-flux assay. Data are presented as Δ fluorescence upon Tl+ addition to media. ATP was present to block the mKATP channel. The baseline Δ fluorescence (Ctrl, set to 100%) was 24.7 ± 2.3 and 24.9 ± 4.4 arbitrary units in WT (white bars) and Slo1−/− (gray bars), respectively. Experimental conditions are listed below the x-axis. Data are means ±SEM, with N for each group listed in parentheses at the base off each bar. Each N represents an independent mitochondrial preparation. Control, ATP and ATP + Pax data are replicated between panels A and B for comparison *p < 0.05 vs. ATP, †p < 0.05 vs. ATP + NS compound (ANOVA).

SLO1 dependent protection in intact hearts requires cardiac neuronal function

So far, our data show that SLO1 is not required for protection by NS1619 and NS11021 in isolated cardiomyocytes (Fig. 3), but is required for protection by these compounds in the intact heart (Figs. 1 and 2). This suggests that in the intact heart, protection occurs via SLO1 in a mechanism that is non-cardiomyocyte autonomous. Several non-myocyte cardioprotective mechanisms exist, including remote preconditioning which is thought to comprise a neuronal signaling component (Hausenloy & Yellon, 2008). We therefore sought to investigate a role for SLO1 in cardioprotection via non myocyte signaling, with a focus on intrinsic cardiac neurons. Importantly, the ex-vivo heart preparation used herein is denervated, such that surviving neurons are not central but rather they are intrinsic to the heart. In addition SLO1 channel activity has been described in cardiac neurons (Franciolini et al., 2001) and Purkinje fibers  (Callewaert, Vereecke & Carmeliet, 1986), where it plays a role in regulating heart rate (Wojtovich et al., 2011; Imlach et al., 2010).

Figure 5 Blocking cardiac neuronal function blocks SLO1 dependent protection in the intact heart.

(A, B): Perfused FVB mouse hearts from WT animals were subjected to IR injury. Where indicated, hearts were treated with NS1619 (5 µM), atpenin A5 (AA5, 10 nM), or hexamethonium (Hex, 500 µM) as detailed in the methods. Left-ventricular function was determined as rate pressure product (RPP; heart rate x left ventricular developed pressure) and expressed as a percent of initial value. Data are split across two panels for clarity. Statistical significance between groups was determined using a mixed-effects model. *p < 0.05 vs. IR alone, #p < 0.05 vs. NS1619 + IR. (C): Following IR protocols, hearts were sliced, stained with TTC and fixed to delineate live (red) from dead or infarcted tissue (white). Infarcts were quantified by planimetry and expressed as a percent of the area at risk (100% in this global ischemia model). Images above the graph show representative slices from each group. Within each group, individual values are on the left, and means ±SEM on the right (N = 8 for IR, 7 for AA5 + IR, and 6 for all other groups). *p < 0.05 vs. IR alone, #p < 0.05 vs. NS1619 + IR.

Figure 5 shows that blockade of neuronal function with the synaptic transmission inhibitor hexamethonium, prevented cardioprotection (reduction in infarct size) by NS1619 in intact hearts. This effect was specific to SLO1 and NS1619, since hexamethonium had no effect on protection afforded by the mitochondrial KATP channel activator atpenin A5 (Wojtovich & Brookes, 2009). These data also indicate that NS1619 protects via a mechanism that is distinct from mitochondrial KATP channel opening.

To further investigate the role of cardiac neurons in protection afforded by SLO1 activators, we assessed neuronal status by histochemical staining for acetylcholinesterase activity (AChE). No differences in AChE staining were found between control, IR, and IR + NS1619 hearts, suggesting the acute time frame of IR injury in this model was not sufficient to precipitate neuronal loss (Fig. 6A–C). We therefore hypothesized that a more direct test of cardiac neuronal function might be necessary to assess neuronal protection in this acute model. Hence, the response of heart rate to nicotine (which induces transient tachycardia via nicotinic Ach receptors (Tosaka et al., 2003; Haass & Kubler, 1997)) was measured after IR injury. As seen in Fig. 6D, no response to nicotine was seen in control hearts subjected to IR, indicating loss of neuronal function. However, hearts protected with NS1619 exhibited a robust post-IR nicotine response, and this effect was blocked by pre-treatment with hexamethonium, indicating a preservation of nicotine-sensitive intrinsic cardiac neurons. Notably, hearts protected with the mKATP opener atpenin A5 did not exhibit a preserved nicotine response. This is consistent with the data in Fig. 4 showing that even though atpenin A5 was protective, this cardioprotection was insensitive to hexamethonium. Overall these data indicate that both mKATP and SLO1 channels can protect the heart (Figs. 1, 2 and 5), but they do so via distinct mechanisms, the latter requiring cardiac neuron function.

Figure 6 Histochemical staining and functional activity of cardiac neurons.

Perfused FVB mouse hearts from WT animals were subjected to IR injury. Where indicated, hearts were treated with NS1619 (5 µM), atpenin A5 (AA5, 10 nM), hexamethonium (Hex, 500 µM), or combinations of NS1619 + Hex or AA5 + Hex, as detailed in the methods. Panels A–C show representative total heart histochemical acetylcholinesterase (AChE) staining following (A) control perfusion, (B) IR injury and (C) IR with NS1619 pre-treatment. Dorsal side of the heart is shown, with magnifications of boxed areas shown in right panels. (D): Functional activity of cardiac neurons was determined following the six IR protocols used in Fig. 3, via the injection of nicotine (100 µM) for 5 min. The peak heart rate increase was found at 2 min, followed by desensitization and subsequent decrease in heart rate. Data are means ±SEM. N for each group is shown in parentheses to the right of the legend. Each N represents an individual perfused heart. *p < 0.05 vs. IR (ANOVA).

Figure 7 Working model of SLO1 (in)dependent mechanisms of action in NS1619/NS11021 cardioprotection.

(A): In isolated cardiomyocytes the NS compounds induce cardioprotection independent of SLO1. The ‘target’ remains unclear but may include inhibition of respiration or the uncoupling of oxidative phosphorylation (see text for details). This mechanism is compatible with the lack of evidence for SLO1 in cardiomyocytes. (B): In the whole heart, cardioprotection by NS1619/NS11021 is dependent on SLO1 and is likely mediated via a cell non-autonomous mechanism, involving intrinsic cardiac neurons. A barrier is hypothesized to exist, possibly accounting for the inability of NS compounds to recruit the nonspecific target responsible for protection in myocytes. This explains the lack of ability of NS1619/NS11021 to protect the intact Slo1−/− heart. HEX = hexamethonium.

Discussion

Together, the current data suggest that the SLO1 openers NS1619 and NS11021 act to preserve intrinsic cardiac neuronal function in IR injury, in a Slo1 dependent manner, and this leads to protection of myocardium. As such, the function of cardiomyocytes in IR injury can be preserved by activation of a non-cardiomyocyte-autonomous signaling mechanism. From a pharmaco-genomic perspective, this is the first study to show that cardioprotection in the intact heart by the putative SLO1 activators NS1619 and NS11021 actually requires the Slo1 gene product (Figs. 1 and 2).

Although the chemical structures of NS1619 and NS11021 are quite diverse, an inactive analog of NS11021 (NS13558) does not elicit cardioprotection  (Bentzen et al., 2010), suggesting these molecules bind to a specific target in the whole heart, presumably SLO1. While the mitochondrial KATP channel would appear to be an obvious alternative target for the NS compounds, protection by NS1619 is not blocked by the mKATP blocker 5-HD (Cao et al., 2005), suggesting this is not the case. Rather, the mechanism of protection by SLO1 openers is distinct from that afforded by activation of cardiac mKATP channels (e.g., by diazoxide or atpenin A5), and suggests that non-myocyte and possibly non-mitochondrial K+ channels are also important in determining responses to ischemia.

The data in Figs. 5 and 6 suggest that cardioprotection by SLO1 activators proceeds via intrinsic cardiac neurons, with preservation of neuronal function during IR injury leading to improved cardiac functional recovery, decreased infarct size (Fig. 5), and the post-IR ability to mount a tachycardic response to nicotine (Fig. 6D). This nicotine response requires both intact cardiac neurons and the ability of myocytes to respond to a tachycardic signal. Notably, hearts protected by mKATP agonist atpenin A5 were fully competent at the contractile level post-IR (Fig. 5), but failed to respond to nicotine. This suggests a post-IR defect in a non-cardiomyocyte cell-type, presumably neurons since this is the only other cell type known to be involved in the nicotine response. Post-IR in NS1619 treated hearts responded to nicotine in a manner that was blocked by hexamethonium (Fig. 6D), and the preservation of contractile function by NS1619 was also blocked by hexamethonium (Fig. 5).

Overall, the most logical explanation for these data is that NS1619 protection proceeds via SLO1 in cardiac neurons. However, we cannot rule out the possibility that NS1619 acts on other neuronal targets. Unfortunately this cannot be tested experimentally using Slo1−/− mice, because post-IR Slo1−/− hearts treated with NS1619 are incompetent at the contractile level (Figs. 1 and 2), so they cannot respond to nicotine anyway, regardless of their neuronal status. Clearly, cardiomyocyte- or neuron-specific Slo1−/− animals would be of enormous benefit in further elucidating the protective mechanism of the NS compounds in future.

While a role for afferent or efferent neurons (or indeed residual synaptic bulbs from central neurons) cannot be completely ruled out (Wenk & McCleskey, 2007), it should be emphasized that the Langendorff perfused heart preparation is denervated upon removal of the heart from the thorax, and thus the neurons of interest to this study are not afferent or efferent to the central system, but are intrinsic to the heart. The presence of ganglionated plexuses within the heart is well documented (Yuan et al., 1994), and a number of important signaling roles have been assigned to such neurons, including cardiac responses to nitric oxide (Armour et al., 1995), endothelin (Armour, 1996), histamine  (Armour, 1996), angiotensin II (Horackova & Armour, 1997), α/β adrenergic agonists  (Armour, 1997), and nicotine (Murphy et al., 1994). Although the intrinsic cardiac neuronal system has been proposed to play a role in cardiac responses to ischemia  (Armour, 1999), the current study is among the first to directly demonstrate this concept.

The activity of intrinsic cardiac neurons is increased during ischemia  (Foreman et al., 2000). Furthermore, SLO1 in neurons regulates neurotransmitter release  (Wang et al., 2001), and the loss of SLO1 results in an increased duration of release  (Wang et al., 2001). Thus, activating SLO1 with NS1619 or NS11021 may hyperpolarize neurons, limiting neurotransmitter release and minimizing the neuronal depolarization and excitotoxicity associated with ischemic events (Gribkoff et al., 2001). In simple terms, the NS compounds may prevent excitotoxicity. The ability to modulate cardiac neuronal activity in this manner is a promising therapeutic avenue for ischemic injury (Armour, 1999). Recently, several non-myocyte cell types in the heart have emerged as key players in the response to ischemic injury, including cardiac fibroblasts (Kawaguchi et al., 2011), cardiac mast cells (Reid et al., 2011; Koda et al., 2010), and cardiac stem/progenitor cells (Shah et al., 2011). At this stage, it is not clear whether the cardioprotection observed herein represents a direct effect of cardiac neurons on cardiomyocytes, and the potential involvement of additional intermediary cell types in transmitting the protective signal cannot be excluded.

The precise sub-cellular location of SLO1 required for protection by NS1619 and NS11021 remains unclear. Early work on the role of SLO1 in cardioprotection found no expression on the surface of cardiomyocytes (Xu et al., 2002). When single-channel recordings of a Ca2+-dependent K+ channel were made in human glioma cell mitochondria (Siemen et al., 1999) a consensus emerged that the cardioprotective effect of SLO1 activators was likely due to a SLO1 channel in cardiomyocyte mitochondria (Xu et al., 2002). Observations that the SLO channel blocker paxilline abrogates protection by NS1619 and NS11021 (Bentzen et al., 2009; Xu et al., 2002), as well as the protection afforded by APC (Wojtovich et al., 2011), are often invoked as proof for the existence of a “cardiac mitochondrial KCa channel”, and this naming convention is consistent with the fact that mammalian SLO1 is a Ca2+ sensitive isoform of SLO (Salkoff et al., 2006). Further support comes from immunological studies, in which antibodies directed against the canonical SLO1 (110–130 kDa) detect the protein in mitochondrial fractions (120, 80, and 55 kDa) (Xu et al., 2002; Fretwell & Dickenson, 2009; Wang et al., 2004; Singh, Stefani & Toro, 2012).

More recently however, the consensus view that SLO1 is a mitochondrial KCa channel important for cardioprotection has been challenged by our observation that SLO1 is dispensable for APC and IPC (Wojtovich et al., 2011). Furthermore, although the SLO auxiliary subunit β4 (Fretwell & Dickenson, 2009; Piwonska et al., 2008; Skalska et al., 2009; Ohya et al., 2005) has been found in mitochondria, this subunit renders SLO1 resistant to charybdotoxin and iberiotoxin – toxins that block the effects of NS1619 on mitochondria  (Meera, Wallner & Toro, 2000). We have also shown that in the model organism C. elegans, the SLO2 isoform of the channel underlies both APC and mitochondrial K+ flux (Wojtovich et al., 2011). While studies are ongoing to confirm the identity of the mitochondrial BK channel, it seems clear that the simple picture of a SLO1 channel in cardiomyocyte mitochondria as the underlying mechanism of cardioprotection by APC or NS compounds is untenable. Indeed, the data in Fig. 3 show that NS compounds elicited K+ transport in isolated mitochondria independent of SLO1, adding to the body of evidence for off-target effects of these compounds at the mitochondrial level.

Subsequently, the question arises as to the mechanism by which the SLO1 openers NS1619 and NS11021 can elicit protection in isolated cardiomyocytes (Fretwell & Dickenson, 2009; Chmielewska & Malinska, 2011), in a Slo1 independent manner (Fig. 3). It seems reasonable to suggest that the well-documented off-target effects of these molecules (reinforced by data in Fig. 4) may underlie such protection (Aldakkak et al., 2010; Cancherini, Queliconi & Kowaltowski, 2007; Debska et al., 2003; Kicinska & Szewczyk, 2004). Both NS1619 and NS11021 have been reported to inhibit respiration (Cancherini, Queliconi & Kowaltowski, 2007; Kicinska & Szewczyk, 2004), promote mitochondrial depolarization (Aon et al., 2010; Cancherini, Queliconi & Kowaltowski, 2007; Debska et al., 2003; Kicinska & Szewczyk, 2004) and cause mitochondrial swelling independent of K+ (Cancherini, Queliconi & Kowaltowski, 2007; Bednarczyk, Barker & Halestrap, 2008) suggesting they can impact non-specific ion transport mechanisms (Aldakkak et al., 2010; Cancherini, Queliconi & Kowaltowski, 2007). NS1619 has also been shown to increase mitochondrial ROS generation without depolarization (Heinen et al., 2007), and a role for small amounts of ROS in cardioprotective signaling is well known (Vanden Hoek et al., 1998). Alternatively, NS1619 and NS11021 may protect cardiomyocytes via non-mitochondrial targets that remain to be identified.

Mitochondrial ion channels such as the mKATP and SLO2 remain important mediators of IPC and APC cardioprotective signaling (Facundo, Fornazari & Kowaltowski, 2006; Wojtovich et al., 2011; Murphy & Steenbergen, 2007; Garlid et al., 2009; O’Rourke, 2004), and it should be emphasized that the data presented herein do not preclude the presence of a SLO1 channel in the mitochondria. Slo1 has many splice variants  (Salkoff et al., 2006) and it is possible that such variants may be present in the mitochondrion, but these channels are either not the primary target or lack the binding site necessary to respond to NS1619 and NS11021. Furthermore, the Slo1−/− mouse used in these studies has a deletion of Exon 1, which encodes the S0 domain required for channel function (Meredith et al., 2004; Zarei et al., 2001; Zarei et al., 2004), making it unlikely that a functional splice variant is present in the knockout. Further studies are needed to determine the nature of any SLO1 in mitochondria, or to investigate the intriguing notion that SLO1 may reside in the mitochondria of intrinsic cardiac neurons.

A surprising paradox emerges when comparing the whole heart and cardiomyocyte effects of NS1619 and NS11021 (Figs. 2 and 3). Specifically, if these compounds are capable of protecting isolated cardiomyocytes in a Slo1 independent manner, why is this mechanism of protection not accessible in whole hearts? If it were, the compounds should protect the Slo1−/− heart. One possibility may be the different experimental systems, or that the isolated myocytes does not express SLO1, which may promote nonspecific actions of these compounds. Another possibility is that a barrier exists in the intact heart (possibly a vascular diffusional barrier) which prevents NS1619 and NS11021 from activating the protective pathway observed in cells. Notably, attempts to overcome this limitation by using high concentrations of NS1619 (30 µM) in the intact heart led to an exacerbation of IR injury (data not shown), which may be related to reports that high concentrations of this compound can depolarize mitochondria (Cancherini, Queliconi & Kowaltowski, 2007). Notably, a recent review article stated that “In whole heart experiments the effective concentration of NS1619 is actually lower due to limited diffusion of the drug to its site of action, and thus, it promotes cardioprotection instead of mitochondrial damage and heart stress.” (Singh, Stefani & Toro, 2012)

Clearly in the intact heart, something prevents NS1619 and NS11021 from protecting myocytes directly, such that the only avenue of protection available to these molecules is via SLO1, and knocking out the channel abrogates this protective mechanism. A schematic outlining the different actions of NS1619 and NS11021 in isolated myocytes vs. intact hearts is shown in Fig. 7, with the exact nature of the protective signal initiated at the cardiac neuronal level by SLO1 openers still to be elucidated.

In conclusion, the SLO1 channel is a cardioprotective target of NS1619 and NS11021, and these molecules are important tools for assessing SLO1 function in the whole heart. The molecules elicit protection in isolated cardiomyocytes that is independent of SLO1, possibly mediated by off-target effects. In contrast at the whole heart level, protection by NS1619 and NS11021 absolutely requires SLO1, and preserves intrinsic cardiac neuronal function. Collectively these studies support the notion that intrinsic cardiac neurons are important mediators of cardioprotection, and may be a therapeutic target (Hausenloy & Yellon, 2008). Abbreviations

NS1619 1,3-Dihydro-1-[2-hydroxy-5-(trifluoromethyl)phenyl]-5-(trifluoromethyl)-2H-benzimidazol-2-one

NS11021 1-(3,5-bis-trifluoromethyl-phenyl)-3-[4-bromo-2-(1H-tetrazol-5-yl)-phenyl]-thiourea

BTC-AM Benzothiazole coumarin acetyoxymethyl ester

BK large-conductance “big” K+ channel

mKATP mitochondrial ATP-sensitive K+ channel

TTC Tetrazolium chloride

WT Wild-type

We thank James Melvin (NIH) and Richard Aldrich (University of Texas at Austin) for the Slo1−/− mice.

Additional Information and Declarations

Competing Interests

Author Contributions

Animal Ethics

Morten Grunnet is a former employee of Neurosearch A/S. Morten Grunnet is a current employee of Lundbeck A/S. Paul Brookes is an Academic Editor for PeerJ.

Andrew P. Wojtovich conceived and designed the experiments, performed the experiments, analyzed the data, wrote the paper.

Sergiy M. Nadtochiy performed the experiments, analyzed the data.

William R. Urciuoli performed the experiments.

Charles O. Smith analyzed the data.

Morten Grunnet analyzed the data, contributed reagents/materials/analysis tools.

Keith Nehrke conceived and designed the experiments, analyzed the data, contributed reagents/materials/analysis tools.

Paul S. Brookes conceived and designed the experiments, analyzed the data, contributed reagents/materials/analysis tools, wrote the paper.

The following information was supplied relating to ethical approvals (i.e. approving body and any reference numbers):

University of Rochester UCAR (University Committee on Animal Resources).

All studies performed under protocol # UCAR 2010-030

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
