# Peer review of "A non-cardiomyocyte autonomous mechanism of cardioprotection involving the SLO1 BK channel"

_PeerJ, doi:10.7717/peerj.48_

## Round 0.1 · original submission · Minor Revisions

1) Using of different ischemia duritions should be validated
2) Please provide original recording traces for some results.
.

Reviewer 1 ·

Basic reporting

Opening the BKCa channels leads to protection against ischemia and reperfusion (IR) injury. Recent studies have reported that in cardiomyocytes, mitochondria, specifically K channels in mitochondria are partly involved in the mediation of this protection. However, the true pharmacological effects of agonists for KCa2+ channels in ventricular myocytes have stirred debate about the actual target of these agonists. Adding to this uncertainty, Wojtovich et al. in the current study have brought forth a compelling hypothesis that the cardioprotection against IR injury following stimulation of the KCa channels is mediated in the intact heart via a SLO1-dependent mechanism that involves intrinsic cardiac neurons. The rationale is well articulated and provides a compelling reason for the execution of the study.

Experimental design

The experiments are well laid out, with very strong supportive data and a conclusion that is highly helpful of the overarching proposed hypothesis. The entire experimental approach was described clearly with a logical flow tot he sequence of proposed experiments to support the objectives of the project.

Validity of the findings

The results as described in the manuscript are clear and the legends highly descriptive of the figures. The results support the hypothesis put forth. They (results) clearly provides a novel insight into the role of KCa channels in cardioprotection against IR injury. This is analogous with the concept of remote pre- post-conditioning. In this regard, the authors proposed a well-thought out scheme that describes how a "remote cardioportection" is manifested, with intrinsic cardiac neurons being part of the upstream instigators of KCa-mediated protection.

Additional comments

Although the manuscript was well planned, there are minor comments that require the attention of the authors and clarification of these concerns could enhance the overall readability of the manuscript.

On page 4, there seems to be two ischemia times, a 35 vs. 40 min. It is unclear why the two different ischemic times. The authors should be cognizant of the fact that the duration of ischemia after the ideal 30 min ischemic time is significant in inflicting damage to cells and organs.

Page 5, “To assess neuronal survival…” Insert comma (,) after "survival". In the perfused isolated heart model, were cardiac functions recorded. If so, what was the effect IR ± NS on cardiac systolic and diastolic contractility (please provide the original data traces). A prolonged ischemia (35-40 min long) could lead to exaggerated diastolic contracture and the efficacy of the NS drugs could be verified by their abilities to attenuate the diastolic LVP or end diastolic LVP rise.

Page 11, fourth paragraph add “to” after ability to “The ability….”

Page 14, end of third paragraph, what is 68 after target?

·

Basic reporting

Page 9. “In addition SLO1 is known to be expressed in cardiac neurons and Purkinje fibers, where it plays a role in regulating heart rate [15,5]”
The evidence of SLO1 expression in cardiac neurons seems to be rather indirect, also in the cited literatures 5 and 15.

Minor points:

Page 13. “… S- domain required for channel function”
Meaning “S0 domain”?

Page 14. “… may be a therapeutic target 68.”
“68” should be a citation number.

Experimental design

No comments.

Validity of the findings

Page 11. “It should be emphasized that the Langendorff perfused heart preparation is denervated upon removal of the heart from the thorax, and thus the neurons of interest to this study are not afferent or efferent to the central system, but are intrinsic to the heart.”
Considering that the synaptic transmission is intact also in dissected nerves as observed in a muscle-nerve preparation (J Neurosci Methods 159(2):244-51, 2007), it seems that an involvement of afferent/efferent neurons cannot be completely ruled out.

Page 14. “Clearly in the intact heart, something prevents NS1619 and NS11021 from protecting myocytes directly”
NS1619 and NS11021 seem to be rather small molecules, and membrane-permeable as the authors depict in Fig. 6. Considering that capillaries and cardiomyocytes are closely located, it would be difficult to assume a physical barrier between them. There could be another possibility that the discrepancy between the ex-vivo heart and the isolated cardiomyocyte results is due to the difference in the experimental systems.

Additional comments

This research is very intriguing in that it implies an active role of the cardiac neuronal system in responses to ischemia. Further studies regarding the underlying mechanism are highly expected in future.

---

## Round 0.2 · Minor Revisions

Please make any revisions following the comments from the Reviewer #1.

Reviewer 1 ·

Basic reporting

No comment.

Experimental design

Well executed experimental designs and protocols.

Validity of the findings

Highly supportive of the objective of the study.

Additional comments

The revised article entitled “A non-cardiomyocyte autonomous mechanism of cardioprotection involving the SLO1 BK channel” by Wojtovitch et al. has satisfactorily addressed by the authors. My concerns of the first submission have been satisfactorily addressed in the revision and the article is highly improved. There are few minor concerns to be addressed

In the Material and Method section, having periods/full stop after every time a time value or volume is given, e.g. 40 min. is distracting. Remove all such punctuations.

Page 6 first paragraph lines 5-6 “…then 1 hr ischemia buffer, 30 min normoxic buffer.” Add to this sentence “followed by 30 min normoxic buffer.”

Page 13 under Discussion 2nd paragraph lines 5-7, the authors should also acknowledge that other studies have reported that NS1619 induces ROS production without change in mitochondrial membrane potential and that its protection against ischemia reperfusion injury in the isolated perfused heart model may be mediated in part by mitochondrial-dependent reactive oxygen species mechanism.

Page 6 3rd paragraph line 8 “…an functional splice…” Change “an” to “a”.

·

Basic reporting

Reporting is tactful and accurate.

Experimental design

No Comments

Validity of the findings

The discussion is well balanced.

Additional comments

This research is very intriguing in that it implies an active role of the cardiac neuronal system in responses to ischemia. I am looking forward to having new finding regarding this topic.

---

## Round 0.3 · accepted · Accept

Congratulations on your excellent work!